# Towards Improved Human *In Vitro* Models for Cardiac Arrhythmia: Disease Mechanisms, Treatment, and Models of Atrial Fibrillation

**DOI:** 10.3390/biomedicines11092355

**Published:** 2023-08-23

**Authors:** Carla Cofiño-Fabres, Robert Passier, Verena Schwach

**Affiliations:** 1Department of Applied Stem Cell Technologies, TechMed Centre, University of Twente, Drienerlolaan 5, 7500 AE Enschede, The Netherlands; c.cofinofabres@utwente.nl; 2Department of Anatomy and Embryology, Leiden University Medical Centre, 2300 RC Leiden, The Netherlands

**Keywords:** HPSCs, *in vitro* models arrhythmia, cardiac disease modeling

## Abstract

Heart rhythm disorders, arrhythmias, place a huge economic burden on society and have a large impact on the quality of life of a vast number of people. Arrhythmias can have genetic causes but primarily arise from heart tissue remodeling during aging or heart disease. As current therapies do not address the causes of arrhythmias but only manage the symptoms, it is of paramount importance to generate innovative test models and platforms for gaining knowledge about the underlying disease mechanisms which are compatible with drug screening. In this review, we outline the most important features of atrial fibrillation (AFib), the most common cardiac arrhythmia. We will discuss the epidemiology, risk factors, underlying causes, and present therapies of AFib, as well as the shortcomings and opportunities of current models for cardiac arrhythmia, including animal models, *in silico* and *in vitro* models utilizing human pluripotent stem cell (hPSC)-derived cardiomyocytes.

## 1. Introduction

Cardiovascular diseases cause 32% (about 18 million) of global deaths yearly, according to the World Health Organization [1], and are the leading cause of death. Cardiovascular disease can be classified into diseases of the heart muscle, the valves, and vessels. Heart rhythm disorders, arrhythmias, are irregular heartbeats and form one very common type of heart disease. Arrhythmias can be subdivided into those with slower (bradyarrhythmias) or faster (tachyarrhythmias) heart rates than a physiological resting heart rate (60–100 beats per minute (BPM)). Atrial fibrillation (AFib), the most common arrhythmia, is characterized by uncoordinated electrical activity and contraction of the atria, which causes a rapid and irregular heartbeat higher than 100 BPM.

Over time, AFib is associated with a doubling in all-cause mortality, a five-fold increase in stroke, an accelerated development of heart failure, thromboembolic events, and a substantially poorer quality of life [2]. Since the duration of a specific AFib episode can drastically vary between patients, AFib is usually divided into three main categories: (1) paroxysmal AFib, when episodes last less than 7 days and spontaneously convert to normal sinus rhythm, (2) persistent AFib when episodes last more than 7 days, or (3) permanent AFib, a chronic arrhythmia [3].

According to a 2019 global burden of disease (GBD) study, AFib affects an estimated 59.7 million people (30.28 million men and 29.41 million women) worldwide, with a rising incidence and precedence, leading to almost five million new cases of AFib annually. Incidence, prevalence, and mortality of AFib are strongly dependent on age and gender, as well as ethnicity and race (Figure 1). Incidence is significantly higher in developed regions compared to developing countries. The highest prevalence estimated was for North America, while the lowest was found in the Sub-Saharian Africa region (2364.9 vs. 134.31 cases per 100,000 people). However, the prevalence of AFib may be underestimated because of the large number of asymptomatic individuals. Over the next 2 decades, AFib is projected to increase in an exponential manner, perchance doubling in prevalence, which is essentially due to the aging of the population, but also to the improved ability to treat other diseases. In the particular case of Europe, the number of patients with AFib is estimated to be between 14 and 17 million in 2030. Thus, AFib is currently identified as one of the biggest public health problems and even a growing epidemic, placing a large burden on both social and economic levels [4,5,6,7,8,9].

Many studies have reported several risk factors associated with AFib, for example, increasing age, ethnicity, and gender (Figure 2). Common cardiovascular risk factors are thought to promote AFib by increasing atrial pressure and/or by causing atrial dilation [2]. Although the majority of AFib subjects develop arrhythmic behavior in the context of preexisting cardiovascular disease, genetic factors may play an important role in the onset and maintenance of the disease [10]. Many ion channel genes have been re-sequenced, and a cascade of new mutations associated with AFib has been identified (Figure 2). For example, gain-of-function mutations are predicted to promote repolarization, shorten the atrial action potential and facilitate re-entry [11], while loss-of-function mutations seem to lead to prolongation of the atrial action potential, characteristic of atrial remodeling [12]. Genome-wide association studies have also provided evidence implicating common genetic variants in the pathogenesis of AFib. Therefore, understanding the mechanisms underlying the complex relationship between AFib and its associated multiple risk factors is crucial for the development of AFib models predictive for the human condition.

Arrhythmias, including AFib, are leading causes of morbidity and mortality, predominantly through increased risk of stroke, heart failure, and other heart-related complications [29]. Modeling AFib has primarily focused on different *in vivo* animal models to understand the underlying causes of the disease and the development of new treatments [30]. However, animals do not recapitulate human physiology because of species differences in cardiac physiology, such as expression, distribution, and function of contractile proteins or ion channels [31,32]. Hence, there is an urgent need for predictive modeling of AFib, capable of dynamically mimicking and incorporating various known cardiac disease features and risk factors. In recent years, the realization of efficient protocols for the differentiation of cardiovascular cellular subtypes from human pluripotent stem cells (hPSCs) has made a significant contribution to the development of personalized cardiac disease models. HPSC-derived progeny, such as ventricular or atrial cardiomyocytes, have been established in preclinical disease modeling, drug testing, as well as advancing new therapy [33]. In this review, we will describe the current knowledge about the underlying disease mechanisms and therapy of AFib. After an overview of animal models to study arrhythmia, we will dive into an extended discussion on the state-of-the-art *in vitro* models utilizing hPSC-derived cardiomyocytes for AFib modeling. Finally, we will provide our future perspectives on the field.

## 2. Underlying Disease Mechanisms of AFib

### 2.1. Brief Overview of Normal Electrophysiology of Cardiomyocytes

In a healthy heart, cardiomyocytes are electrically coupled together, forming a functional syncytium that allows the rapid propagation of the electrical impulse throughout the cardiac muscle, causing continuous and highly coordinated contractions. Each action potential is initiated by the sinoatrial (SA) node, a specialized group of myocardial cells located in the posterior wall of the right atrium, close to the orifice of the superior vena cava (Figure 3A). Pacemaker cells exhibit automaticity, meaning they spontaneously depolarize to initiate the action potential. Even though SA node cells are not the only conductive cells, these are the ones with the highest rate of depolarization and thus are the primary pacemaker of the heart responsible for establishing the normal sinus rhythm. After being generated in the SA node, the impulse spreads to the adjacent atrial cardiomyocytes through cell-to-cell gap junctions, as well as radially via a specialized internodal pathway called Bachmann’s bundle or the interatrial band. This structure conducts the impulse directly from the right atrium to the left atrium. Ultimately, the wave of depolarization reaches a second group of specialized cells at the bottom of the right atrium, the atrioventricular (AV) node. Because the atria and ventricles are electrically isolated from one another by a circumferential band of fibrous tissue, the only path for impulse propagation is via the AV node. After a brief, intrinsic delay at the AV node, the action potential is propagated quickly down the bundle of His and Purkinje fibers within the ventricular myocardium [34] (Figure 3A). At rest, the cardiomyocyte membrane is more permeable to potassium (K^+^) than sodium (Na^+^) or calcium (Ca^2+^), resulting in a negative electrical gradient with respect to the extracellular environment since potassium has an equilibrium potential of around −80 Mv, the resting membrane potential of cardiomyocytes. With sufficient stimulus, alterations in the permeability to Na^+^ result in a rapid increase in the membrane potential to values of about 40 Mv, creating a net positive electrical gradient (depolarization). Further, changes in the ion permeability to K^+^, Cl^−^, and Ca^2+^ results in the eventual restoration of the negative intracellular environment (Figure 3B). The action potential is linked to the cardiac contraction via excitation-contraction coupling. In the healthy condition, contraction is induced by an influx of Ca^2+^ ions through voltage-dependent L-type channels [35,36,37] after these have been opened by depolarization through the influx of Na^+^. This initial increase in Ca^2+^ in the cytosol triggers the release of Ca^2+^ from the sarcoplasmic reticulum (SR) through Ryanodine receptor 2 (RyR2). The high levels of cytosolic Ca^2+^ trigger contraction. Relaxation is achieved by moving cytosolic Ca^2+^ into the SR via sarco/endoplasmic reticulum Ca^2+^-ATPase (SERCA2a) or through the plasma membrane via the rheogenic Na^+^/Ca^2+^-exchanger (NCX), which transports 1 Ca^2+^ ion outward while moving 3 Na^+^ ions inward.

### 2.2. Trigger and Substrate Cause AFib

The pathophysiology of AFib contains three major components, namely onset, maintenance, and progression toward longer-lasting AFib forms. AFib is initiated as a response to a trigger rapidly exciting the atria in combination with a substrate able to maintain fibrillations. Generally accepted trigger phenomena are rapidly firing ectopic foci, re-entrant waves, and rotors, while substrates include electrical, structural, and autonomic remodeling [38,39].

### 2.3. Triggers: AFib Is Induced by Ectopic Foci, Re-Entry and Rotors

To be initiated, AFib generally requires an initiating electrical stimulus, a ‘trigger.’ Three main sources as triggers for the rapid and uncoordinated atrial activity generated in AFib have been established: (1) rapidly firing ectopic foci with irregular conduction and fibrillatory activity; (2) re-entrant activity, which can be associated with complex multiple re-entrant circuits, or (3) rotors which are formed by functional re-entry spiral waves with irregular patterns and no consistent activation pattern [37] (Figure 4). Triggers often arise from the pulmonary veins [40] (Figure 4A). Atrial ectopic activity arises from triggered activity as a result of diastolic delayed afterdepolarizations (DADs) [3,41,42] or spontaneous extra-systolic activity by early afterdepolarizations (EADs) [43]. DADs are spontaneous depolarizations occurring in cardiomyocytes after the repolarization phase of the action potential, while EADs occur during the repolarization phase [35,44]. EADs or DADs arise because of disturbed calcium homeostasis. If the DAD is large enough, it depolarizes the cell and causes a premature atrial ectopic beat. A series of DADs can cause atrial tachycardia, which can trigger atrial re-entry or, if rapid enough can maintain AFib. Direct evidence for DAD-mediated ectopic activity related to abnormal Ca^2+^ handling has been provided in animal models of AFib [45], as well as in patients with AFib [46]. A high Ca^2^ load in the cytoplasm at the repolarization phase causes depolarization through the NCX and consequently aberrant excitation and impulse formation [41,42]. The high amount of Ca^2+^ in the cytosol can be caused by tachycardia, ischemia, β-adrenergic receptor stimulation, low extracellular potassium concentration [47], or SR Ca^2+^ leaks [41,42]. SR Ca^2+^ leaks can be caused by increased SERCA2a activity [46] and dysregulated RyR_2_ or L-type channels that cause an increase in SR Ca^2+^ load [3,48] (Figure 5).

Re-entry is the phenomenon of activation that can travel and re-excite the tissue it originated from [38]. As a trigger, re-entry is heavily dependent on a supporting substrate, as cardiac tissue cannot be activated again during the refractory period. The likelihood of re-entry in a re-entrant circuit is defined by the distance traveled by the wavefront during the refractory period (wavelength of the circuit). If the wavelength of the circuit is equal to its circumference due to either a short refractory period or low conduction velocity, there will be no excitation gap. Thus, the leading edge of the impulse will trespass the excitable and recovered tissue, establishing the smallest circuit (leading circle) with a continuously depolarized center around a conduction barrier, which can be the pulmonary vein or scar tissue [39,49] (Figure 4B middle).

Rotors are another type of re-entry, where the wave moves in a spiral or curved form instead of in a circle (Figure 4B right). Where the wavefront and wave tail meet, the singularity point is formed. Due to a current-to-load mismatch (see below), the wavefront curvature influences the wavefront conduction velocity. Near the singularity point, the curvature of the wavefront is so strong, and the conduction velocity is consequently so slow that an excitable core of tissue cannot be excited, forming a functional but not refractory block [39,49].

### 2.4. Current-to-Load (Source-Sink) Mismatches Affect Conduction Velocity

One of the mechanisms that maintain re-entry is current source-sink mismatches, where the action potential propagates from a source tissue with depolarizing current towards the sink tissue that will be depolarized (Figure 6A). Current-to-load mismatches at boundaries between inhomogeneous tissue volumes change the conduction velocity and may lead to the onset or maintenance of arrhythmia. Conduction velocity in a 2D isotropic medium can be described as: $\theta =\theta_{0}+D\rho$ [56]. Where theta is the conduction velocity, theta_0_ is the conduction velocity of a non-curved wavefront, and ρ denotes wavefront curvature. D is a coefficient equal to 1/*C*_m_*S*_v_*R*_i_ where *C*_m_ is the specific membrane capacitance, *S*_v_ is the cell surface-to-volume ratio, and *R*_i_ is the intracellular resistivity. Thus, if the source and sink are the same size (ρ remains equal), the signal will propagate without a change in conduction velocity. When the volume of the sink tissue is smaller compared to the source tissue (positive ρ, resulting in a concave wavefront), the electrical propagation will continue with increased velocity as there is sufficient electrical current available to depolarize the sink tissue. However, when the source tissue is smaller than the sink (negative ρ, convex wavefront), the conduction velocity will slow or even be blocked [57,58,59] (Figure 6B). This decreased conduction velocity predisposes to arrhythmia.

### 2.5. Substrate: AFib Is Maintained via Electrical, Structural, or Autonomic Remodeling of Atrial Tissue

A vulnerable atrial substrate will enhance susceptibility for AFib induction, maintenance, and progression. Atrial remodeling, regarding electrical and structural properties as well as autonomic innervation, constitutes a substrate for AFib (Figure 5).

Electrical remodeling in AFib consists of several components; those identified so far are (1) decreased L-type Ca^2+^ channel current, (2) increased K^+^ currents, and (3) abnormalities in gap junction ion channel functions [61]. In the first case, the overload of Ca^2+^ often present in AFib causes restoration of homeostasis by activation of the NFAT system, which inhibits, at the same time, transcription of Ca_v_1.2, a subunit of L-type Ca^2+^ channels. Consequently, the duration of the atrial action potential shortens, leaving the tissue more vulnerable to re-entry [45,62]. Secondly, upregulation of K^+^ currents in AFib, such as the inward rectifying K^+^ current (I_K1_) channel protein K_IR_2.1 or the atrial-specific acetylcholine-activated inward rectifier potassium current (I_KACh_), is associated with shortened action potential duration [37,63]. Moreover, atrial tachypacing has been shown to increase the expression of Ca^2+^ dependent K^+^ (SK) channels and enhance the trafficking of SK channels to the cell membrane, contributing to the reduction in the action potential duration [64,65]. Finally, the electrical coupling between cardiomyocytes is mediated by gap junction ion channels [61]. Overexpression of atrial-specific connexin 40 or ventricular-enriched connexin 43 has been shown to increase conduction velocity, thus preventing the development of persistent AFib in pigs [66]. Therefore, alteration of connexins could potentially lead to conduction velocity abnormalities, contributing to the development of AFib.

Structural remodeling of the atria in the context of AFib mainly involves fibrosis, which makes AFib more persistent and resistant to therapy over time [61]. Cardiac fibrosis involves increased deposition of extracellular matrix components (ECM), including collagen I and fibronectin, mainly produced by myofibroblasts [67]. This fibrotic tissue can form a conduction barrier, where an unexcitable zone at the center of the circuit is produced, being an anchor for re-entry. The border between the fibrotic and non-fibrotic tissue might help stabilize the re-entry. A recent optical mapping study in explanted human hearts showed evidence that re-entry underlies AFib and found that transmural re-entry was stabilized by anatomical structures with local fibrosis [68,69,70]. Additionally, to fibrosis, oxidative stress increases the levels of Reactive Oxygen Species (ROS). High ROS levels can lead to structural remodeling by inducing inflammatory reactions or direct oxidative damage to myofibrils. Oxidative stress damage to mitochondrial DNA can also drive to modulation of Ca^2+^ handling proteins and/or channels, causing a Ca^2+^ overload, leading to electrical remodeling [71,72].

Finally, the autonomous nervous system consists of the sympathetic (fight or flight) and parasympathetic (rest and digest) nervous systems, which mostly act as antagonists to each other. Both subsystems innervate the heart, with the sympathetic (vagal) system increasing and the parasympathetic system decreasing heart rate. Increased firing from both the sympathetic and the parasympathetic systems contribute to AFib [73]. Parasympathetic activation leads to the release of acetylcholine, which increases I_KACh_, thus shortening the action potential duration and supporting re-entry. Sympathetic innervation has been shown to increase in response to rapid atrial pacing and leads to β-adrenergic receptor stimulation. This leads to protein kinase A phosphorylation, which in turn phosphorylates several proteins involved in SR Ca^2+^ loading [74]. Together these effects raise the risk of EAD, DAD, and re-entry [75,76].

### 2.6. AFib Begets AFib

As AFib progresses, it may remodel the substrate and reinforce the vulnerability of the substrate to maintain AFib. This self-reinforcement is known as “AFib begets AFib” and limits the effectiveness of treatments after prolonged AFib [61] (Figure 5). AFib results in rapid shortening of the refractory period as a result of disturbed ion channel activity. The shorter refractory period stabilizes AFib, as indicated by spontaneous AFib maintenance [43,49]. Another contributor to this self-reinforcement is atrial fibrosis. Rapidly firing atrial cardiomyocytes generate substances that make fibroblasts differentiate into collagen-producing myofibroblasts, hence reducing the cardiac impulse propagation homogeneity and inducing irregular rhythm. AFib becomes increasingly persistent and resistant to therapy over time.

## 3. Treatment of AFib

Despite the availability of various therapies, treatments for AFib (such as long-term pharmacology, cardioversion, or surgical intervention by ablation) primarily focus on alleviating the symptoms and have significant limitations, making AFib still a clinical challenge. Cardiac arrhythmias have been traditionally treated by rate or rhythm control.

Rate control therapies focus on the reduction in the rapid ventricular rate (often found in patients with AFib) by the use of negative chronotropic drugs (beta-blockers, calcium channel blockers, cardiac glycosides, or combination therapies) or electrophysiological/surgical interventions. This treatment effectively reduces the symptoms of AFib [77].

Rhythm control therapies aim to restore proper sinus rhythm through drugs (pharmacological cardioversion) or electric currents (electrical cardioversion). After successful cardioversion, patients are generally treated with long-term anti-arrhythmic drug therapy to prevent the recurrence of AFib by blocking the function of ion channels, reducing the excitation of the cardiomyocytes, and prolonging action potential duration [71,77]. Anti-arrhythmic drugs in clinical use (and their target current) are flecainide (I_Na_), propafenone (I_Na_), ibutilide (I_Kr_), vernakalant (multiple ion channel blocker), amiodarone (multiple ion channel blocker), dronedarone (multiple ion channel blocker), quinidine (I_Kr_), disopyramide (I_Na_), solatol (I_Kr_) and dofetilide (I_Kr_) (not available in Europe). Of these anti-arrhythmic agents, potassium ion channel blockers are preferred in the treatment of AFib. The choice of agent is dependent on the persistency of AFib and preexisting heart conditions. The success rates of these drugs in restoring and maintaining the sinus rhythm range from 30 to 50% [78]. Rhythm control, together with rate control, does not improve the clinical outcome of AFib when compared to just rate control. However, rhythm control is still employed for symptom relief [71,79,80].

Rate and rhythm control strategies require a co-treatment with anti-thrombotic drugs to lower the risk of stroke and thromboembolic events [77].

The main drawback to most anti-arrhythmic drugs for the treatment of AFib is non-specificity to the atria that paradoxically leads to life-threatening ventricular ‘proarrhythmias’ [81,82]. Current research efforts, therefore, focus on drugs that mainly target atrial-specific ion channels, I_Kur_ and I_KAch_, and rate-dependent inhibition of I_Na_ [83]. For example, vernakalant inhibits atrial-specific I_Kur_ and I_KAch,_ but also affects I_Kr_, I_to,_ and I_Na_ current. Nevertheless, inhibition of I_Kr_ is significantly lower than that of other drugs, such as flecainide, propafenone, ibutilide, and dofetilide. I_to_ is not exclusive to atria, but its contribution to the atrial action potential is larger compared to the ventricular action potential. Na^+^ channels have a higher affinity for blocking drugs during the action potential. Consequently, inhibition of I_Na_ is more pronounced in fast-beating atria when compared to the ventricles during AFib [82,84]. The progression from paroxysmal to persistent and permanent forms of AFib has pronounced therapeutic implications, with paroxysmal AFib being more amenable to rhythm control therapy [3]. However, most drug treatments have profound side effects and are expensive to use over long periods [85]. Consequently, there has been a shift towards non-pharmacological therapies for cardiac arrhythmias, including electrical cardioversion and controlled destruction of arrhythmia-generating substrate via ablation of tissue close to pulmonary veins, left atrial roof and posterior wall and right interatrial septum [71,79]. A significant weak point in the current treatment guidelines for AFib is that treatments are prescribed based on the persistency of AFib, not on the underlying mechanism of an individual patient [3]. More advanced human models would allow the development of powerful personalized treatments of AFib.

## 4. Models of AFib

Despite all the advantages provided by different types of modeling systems, there are still important limitations that halt their use for direct clinical purposes. The limited possibility of constructing a *de novo* human model of cardiomyocyte electrophysiology and the need for a better understanding of the underlying mechanisms that lead to the onset and maintenance of atrial arrhythmias demand the development of improved and distinct disease models. Nattel, Bourne, and Talajic [86] succinctly outlined the main features of a model system of atrial arrhythmias: (1) The models should be relevant to clinically observed atrial arrhythmias to include structural and functional abnormalities; (2) The atrial fibrillation/flutter should be inducible and able to be maintained chronically to ensure that a putative intervention has indeed terminated the arrhythmia, and it did not occur spontaneously; and (3) it should be possible to restore and maintain normal sinus rhythm.

### 4.1. In Vivo Modeling: Animal Models

AFib has been studied in both large and small animal models with rate-related electrical or atrial-structural remodeling following acute atrial insults and in the presence of autonomic nervous system modulation. Some of the most relevant animal models used for AFib are summarized in Table 1.

Generally, dogs have been extensively used to study AFib. Although the spontaneous occurrence of AFib in dogs, dog models of AFib are experimentally induced, commonly with atrial tachypacing (at a rate of 400–600 bpm), to study their electrophysiology and electrical remodeling. Nevertheless, AFib episodes in dogs are relatively short due to the difficulty to induce and sustain atrial tachycardias [87,88]. Dogs have also been used to study AFib in the context of inflammation (sterile pericarditis), congestive heart failure, atrial ischemia, and the autonomic nervous system [88]. Goats constitute a more robust sustained model for AFib since they are able to better tolerate AFib. Therefore, long-term studies with atrial tachypacing are possible, facilitating the study of structural and electrical remodeling that is associated with persistent AFib [88]. Interestingly, the persistent AFib maintained by the tachycardia-induced electrical remodeling led to the concept of ‘AFib begets AFib’ [87]. Pigs are interesting models to study AFib due to their similarities to human heart size and electrophysiology. Induction of AFib with atrial tachypacing is commonly used, although it also often triggers systolic dysfunction [88]. To prevent the occurrence of systolic dysfunction, atrioventricular node blockade has been used and facilitated the study of self-sustained and persistent AFib for >4 months [89,90].

Structural remodeling has been observed in dogs, sheep, pigs, goats, and humans alike, but differences exist in the nature of remodeling [53]. Despite the fact that different animal models show similar structural remodeling, achieving atrial fibrosis as commonly observed in human AFib is still limiting since tissue fibrosis and degenerative changes are usually related to aging and/or associated heart diseases [37]. Nevertheless, some investigations in AFib dog models have shown that 4 or 6-week rapid atrial pacing promotes ECM synthesis in the atrial tissue and interstitial fibrosis [91,92]. Similarly, chronic atrial dilatation in goat AFib models might create atrial conduction disturbances suggesting the formation of tissue fibrosis [93].

The study of AFib in small animals, such as mice and rabbits, is more challenging due to the large differences in scale and electrophysiology compared to humans. Since mice do not commonly develop AFib, models of AFib require genetic manipulation, after which programmed electrical stimulation is necessary to assess their ability to generate AFib. These models can be taken as interrogations about the functional effects of particular proteins or systems on fibrillatory potential. Interestingly, an investigation with a murine model using optogenetics as a tool to detect and terminate rapid arrhythmia in a safe, repetitive, and shock-free manner holds promise in how these models can be useful to develop potential shock-free therapies for the treatment of AFib [94].

Despite the fact that experimental animal models have been designed to study the pathophysiology of AFib, including molecular basis, ion-current determinants, anatomical features, and macroscopic mechanisms, there are obvious drawbacks when considering their use. First, animal models are not exactly representative of the ionic and molecular basis of AFib in humans as there are significant interspecies differences in cardiac function and also in the type and density of ion channels, pumps, exchangers, and gap junction proteins. Thus, the mechanistically important parameters can be quite different between species, which often complicates the translation to the clinic. As a result, the mechanisms of AFib in animals are not necessarily analogous to those in humans and have not been sufficiently predictive for drug development.

**Table 1 biomedicines-11-02355-t001:** Overview of *in vivo* animal models for AFib.

Animal Model	AFib Model	AFib Promotion	Clinical Causes of AFib
Dog [50,53,87,88]	Paroxysmal and Persistent AFib models	Electrical, structural and autonomic remodeling	Sterile pericarditis, atrial tachycardia remodeling, CHF-related AFib, acute atrial ischemia, atrial volume overload, mitral regurgitation, cesium infusion
Goat [53,88,93,95]	Paroxysmal and Persistent AFib models	Electrical and structural remodeling	Atrial tachycardia remodeling
Pig [53,88,89]	Paroxysmal and Persistent AFib models	Structural remodeling	Atrial tachycardia remodeling
Sheep [88,96,97]	Paroxysmal and Persistent AFib models	Structural and autonomic remodeling	Atrial volume overload, aortopulmonary shunt, atrial tachycardia remodeling
Rabbit [50,88]	Paroxysmal AFib model	Electrical, structural and autonomic remodeling	Atrial volume overload
Transgenic mice [88,98,99]	Atrial conduction abnormalities models	-	Dilated cardiomyopathy, hypertrophic cardiomyopathy, atrial pathology in CHF, atrial tachycardia remodeling

Paroxysmal: AFib of up to 7 days of duration spontaneously terminates; Persistent: AFib lasts >7 days without spontaneous termination; CHF, Congestive Heart Failure; AT, Atrial tachyarrhythmia; HF, Heart failure.

### 4.2. Alternatives to Animal Models

#### 4.2.1. *In Silico* Models

By using *in silico* models, a multitude of models describing atrial cell electrophysiology have been developed over the last few decades for different mammalian species, e.g., rabbit and canine. For human atrial cells, there have been two principal, longstanding models that reconstruct the action potential (AP) using ordinary differential equations (ODEs) based on overlapping experimental data [100,101]. In the absence of human data, both models rely partially on data obtained from other mammals and have slightly different formulations of ionic currents, pumps, exchangers, and others, resulting in divergent behaviors. Nevertheless, outstanding progress with *in silico* modeling has enabled the confirmation of the multiple wavelet hypothesis as a mechanism underlaying AFib [102], as well as the study of the contribution of different ionic currents to AFib. Moreover, 3D realistic human atrial computer models have proved to be a complementary tool for the study of atrial arrhythmias and for the advancement of improved therapies by offering the possibility of testing the effects of different anti-arrhythmic strategies in a human model without irreversible damage to the patients. A clear example is the integration of computer modeling to design a biological integrated cardiac defibrillator and proof in an *in vitro* system its validity to detect and correct arrhythmia [103]. Additionally, these models are also a great platform for studying the contributions of patient-specific structural remodeling to AFib initiation (i.e., fibrosis distribution, fibrotic architecture, and atrial dilatation), and there is even a possibility for *in silico*-guided ablation therapy [104]. Despite these advances, several challenges remain in order to fully exploit the potential of *in silico* modeling. The main challenges include the lack of personalized details, limited availability of experimental data for model validation, limited patient-specific electrophysiological information, and unresolved issues to simulate complex aspects of AFib progression. Future advancements in experimental methodologies, clinical imaging modalities, computational performance, standardization, and the need for clinical evidence to support the routine use of computational models in AFib management will help move this technology forward [105]. For that, the generation of hPSC-derived atrial cardiomyocytes [106,107,108] and the development of atrial tissues [109,110,111] have indicated the potential to model AFib *in vitro* and open new possibilities to generate robust datasets that facilitate *in silico* modeling.

#### 4.2.2. *In Vitro* Modeling Using hPSC-Derived CMs

Due to the clear interspecies variability found in *in vivo* models and the necessity of biological validation in *in silico* models, *in vitro* models with human cardiomyocytes represent a promising alternative to study mechanisms and processes involved in AFib. However, while it is possible to isolate adult heart cells from patients after heart surgery, the difficulty of obtaining such tissue is further compounded by the fact that human cardiomyocytes are non-dividing cells and do not survive long in culture, severely limiting their application for larger-scale studies. Some studies using conditionally immortalized rat or human atrial myocyte (iAM) lines [112,113] are a valuable source of cardiomyocytes in vitro. However, the generation of patient-specific disease models is more challenging, and the associated high investment to produce the lines make them less attractive for disease modeling [113,114]. In addition, the use of living myocardial slices has been an important approach for studying several aspects of AFib over the last years [115,116,117,118]. Since they retain the cellular architecture and connections found in the heart, mechanisms related to electrophysiology, fibrosis, or calcium handling can be studied. In this context, myocardial slices have been utilized to understand the underlying mechanisms of cardiac arrhythmias by studying ectopic foci throughout the atria [16,119], as well as to examine the role of cardiac re-entry involved in tachyarrhythmias [120,121]. Although numerous efforts have been made to maintain them as unaltered as possible once isolated and preserve them for long-term cultivation [115], they are kept in a 2D environment prone to trigger changes in cellular behavior and functionality and potentially affecting the physiological relevance of the observed responses. Therefore it is also necessary to develop other complementary models to this system.

For the last decades, the major source of human cardiomyocytes has come from hPSCs, which have been successfully established to model disease at a single cell level, reliably reproducing the human cardiac electrical phenotype in health and disease. HPSCs can form atrial, ventricular, and pacemaker-like cardiomyocytes *in vitro* by differentiating them in the presence of specific growth factors [122] or small-molecules. For example, atrial- and ventricular-like cardiomyocytes were derived from hPSCs via differential modulation of retinoic acid signaling during differentiation [106,107,108,123]. Initially, cardiomyocyte differentiation protocols generally did not yield well-delineated populations of a single terminal cell type and instead produced mixed populations of atrial, nodal, ventricular, and non-cardiac cells. Therefore, it was crucial to purify cell populations in order to develop cell-type specific disease models in order to avoid any potential contaminating factors. Recently, the use of cell type-specific reporter lines that facilitate the isolation of the subtype of interest by flow cytometry [124] or the use of metabolic selection protocols that enrich the CM population [108] facilitates the retrieval of pure populations of the desired cell type.

Although the cardiomyocytes derived from hPSCs show many typical features of their *in vivo* equivalents, overt symptoms of many diseases at the organ level develop only in the late stages of adulthood. Hence, one important problem with developing physiological and disease models based on hPSCs-derived cardiomyocytes is that these differentiated derivatives present features typical of immature or fetal cardiomyocytes, as evidenced by their spontaneous electrical activity, action potentials with low upstroke velocities, irregular shape, less-developed sarcomeres, metabolism and gene expression [125]. Nevertheless, hPSC-derived atrial cardiomyocytes do possess several specific currents and ion channels, in particular the ultra-rapid delayed-rectifier K^+^ current (I_Kur_), the acetylcholine-activated inward-rectifying K^+^ current (I_KACh_), all of which are largely absent in the ventricles. Therefore, hPSC-derived cardiomyocyte models may represent a valuable platform for the screening of preclinical anti-arrhythmic drugs and cardiotoxicity and modeling cardiac disease. In fact, when the I_Kur_ blocker 4-aminopyridine was tested in hPSC-derived atrial cardiomyocytes, reduced atrial repolarization was observed, while an I_KACh_ blocker (carbachol) reversed carbachol-induced action potential shortening. Since equivalent effects were not demonstrated in ventricular-like cardiomyocytes, atrial cardiomyocytes emerged as a powerful preclinical model for assessing the efficacy and safety of atrial-selective agents [106]. Additionally, one of the first models concerning the induction of re-entrant activity in atrial cells has recently been developed in cell sheets of hPSC-derived atrial cardiomyocytes [109] or ring-like tissues [111]. The resulting rotors induced in the cell sheets were comparable to those observed in previous computer simulations and animal studies, validating the use of such preparations as an AFib disease model. Nakanishi et al. cultured hPSC-derived atrial cardiomyocytes as monolayers in a geometrical narrow-to-wide pattern together with cardiac fibroblasts to mimic the abrupt anatomical transition at the junction of pulmonary veins (PVs) and left atrium (LA) [126]. Under high-frequency field stimulation, the narrow isthmus model and cell heterogeneity provoked impaired electrical conduction [126]. One of the major limitations presented is the absence of spiral waves/rotors (non-induction of AFib). It can be argued that monolayer models may not be applicable to high-throughput screening strategies, given the spatial requirements to produce re-entrant activity and the need for more sophisticated measurements of wave behavior. Other constraints associated with this 2D model are the lack of cell organization, fiber thickness, extracellular matrix content, and electrophysiological differences between PV and atrial cardiomyocytes when compared to *in vivo*. Moreover, the high non-physiological pacing frequency required to induce conduction disturbances and the electrophysiology of the atrial cardiomyocytes, elucidate their immature phenotype [126].

Approaches to improve the maturation of the hPSC-derived cardiomyocytes include advanced 3D models, such as Engineered Heart Tissues (EHTs) [127] or atrial-ventricular co-cultures systems, the Biowire II [128]. Atrial or ventricular hPSC-cardiomyocytes cultured in an EHT format show a higher gene expression for chamber-specific markers compared to monolayers, suggesting an improvement in maturation [110]. Atrial-like EHTs from hPSC have been successfully formed and exhibit different types of rhythm disorders, being suitable candidates to study AFib *in vitro* [110,111]. Lemme et al. presented atrial-EHTs as a model for AFib, considering that the tissues displayed short action potentials prone to tachyarrhythmias and with a shape similar to APs from patients with persistent AFib. Nevertheless, no further studies were followed to modulate the disorder.

In another model, atrial and ventricular-hPSC-derived CMs were embedded in a collagen-hydrogel to form chamber-specific ring-shaped EHTs, and the resultant tissues recapitulated the expected atrial-ventricular physiological differences in their conduction properties. This was further evidenced by optical mapping analysis, where ventricular-EHTs presented an activation pattern initiated at a primary pacemaker region and activating the rest of the tissue synchronously, and the majority of the atrial-EHTs presented arrhythmias with re-entrant nature and different levels of complexity. Interestingly, these arrhythmias could be converted into a normal rhythm by applying electrical field stimulation or exposing the tissue to anti-arrhythmic agents (Vernakalant and Flecainide). In this study, the utility of an *in vitro* model for AFib is evidenced by showing how a drug-testing platform can be used to study pharmacological cardioversion and rhythm-control strategy of arrhythmias [111]. Chronic intermittent tachypacing by optogenetics in hPSC-derived atrial-EHTs has recently been described as a strategy to induce cardiac dysfunction and study the mechanisms of heart failure and arrhythmogenesis [129]. Interestingly, no contractile dysfunction was observed as a result of the optical tachypacing, but some tissues exhibited an arrhythmic spontaneous beating pattern, and various aspects of AFib remodeling were partly recapitulated, such as a trend to a more negative take-off potential and to an increased action potential amplitude. Further research using different stimulation patterns may be valuable to reveal some insights into the mechanisms of the remodeling process during AFib. Using the Biowire II platform, Zhao et al. reported a co-culture tissue formed of hPSC-derived atrial and ventricular cardiomyocytes cultured between two wires. While AFib was not the goal of the study, it shows its potential, by selectively modulating the tissue response to atrial-specific drugs causing tachycardia, such as serotonin [128].

Successful termination of tachycardia has been achieved by burst pacing (20 Hz), light administration, or drug treatment (Flecainide, JTV-519, and E-4031) [130]. Despite the fact that in this study, the model has been described as a tool for ventricular tachycardia, it sheds some light on how to use chronic interval pacing in models with hPSC-atrial-cardiomyocytes to induce other types of arrhythmia, such as AFib. The atrial-ventricular Biowire II enables the growth of thin and cylindrical tissues with two distinct regions: an atrial and a ventricular end. Two parallel wires allow simultaneous measurement of force and Ca^2+^ transients [128]. Tissue conditioning using chronic electrical stimulation with progressively increasing stimulation rates (up to 6 Hz) promotes cardiac maturation, suggesting that it may be a promising model for selective pharmacology and understanding arrhythmia.

A substantial proportion of AFib patients display no history of AFib-associated disorders, which suggests a hereditary component. Indeed, a familial history of AFib has been linked to a 40% risk increase. Over the last years, a number of AFib-related gene loci and mutations have been identified [10,131]. These comprise several developmental regulators, as well as ion channel genes involved in cardiac action potential generation, suggesting a direct causative role of the latter group. Genome editing technologies, such as CRISPR/Cas9, allow the development of isogenic hPSC disease lines of AFib harboring different genetic variants, including rare mutations observed in familial AFib, as well as common risk alleles. This enables the study of patient-specific disease mechanisms and sets the stage for a pharmaco-genomic screen. Combined targeted genetic engineering with cardiac subtype-specific differentiation of hPSCs was conducted by Marczenke et al. to explore the role of K_v_1.5 channels (encoded by the *KCNA5* gene and responsible for I_Kur_) in atrial hPSCs-cardiomyocytes [132]. In this model, CRISPR/Cas9-mediated mutagenesis of integration-fee hPSCs was employed to generate a functional *KCNA5* knockout. Interestingly, both loss and gain-of-function mutations in *KCNA5* have been shown to cause familial AFib [12]. This seeming paradox is explained by the fact that AFib may be promoted via distinct mechanisms, namely, triggered activity and electrical re-entry. While the latter tends to be favored by action potential shortening, the likelihood of the former becomes increased by AP prolongation. Therefore, in addition to their contribution to drug development and discovery, gene-edited stem cell models can serve as a valuable tool for identifying the variables, such as specific SNPs and drugs under comparison, that can optimize the design of clinical trials. By introducing disease-associated genetic variants into hPSCs, it is possible to assess the impact of these variants on cellular responses to drug treatment, thereby increasing the likelihood of designing a clinical trial “in a dish” that is more likely to yield positive results. For example, a pair of anti-arrhythmic drugs (AADs) demonstrating highly variable efficacy in gene-edited hPSC models harboring a risk allele at 4q25 (i.e., a locus associated with *PITX2*) could inspire a more cost-effective clinical trial for this relatively large AFib sub-population with a high likelihood of producing statistically significant outcomes.

## 5. Conclusions and Future Perspectives

AFib remains the most common sustained arrhythmia linked to several serious comorbidities, such as heart failure and stroke, which makes it one of the biggest public health problems. Additionally, currently, available treatments for AFib, namely long-term treatment with anti-arrhythmic drugs, cardioversion, or surgical intervention by ablation, solely rely on treating symptoms, including control of the heartbeat, prevention of blood clotting and rhythm control, thus presenting noticeable limitations and making AFib still a clinical challenge. As a result, there is a high unmet need for groundbreaking predictive models of human AFib, which would allow for the exploration of the disease’s underlying mechanisms and streamline the process of high-throughput drug screening to discover new therapeutic approaches. Despite all advantages provided by different types of modeling systems, there are still constrains that halt their use for direct clinical purposes. Even though animal models have provided an initial understanding of the mechanisms that trigger and maintain AFib, these display significant morphological and physiological features which differ from those found in humans, thus hindering their use. On the other hand, *in silico*, and *in vitro* modeling approaches have proven to be pivotal tools in the progress of AFib treatment. Nonetheless, these models still present restrictions regarding development and subsequent use as a preclinical model. HPSC-derived cardiomyocytes usually exhibit an immature phenotype, a major limitation for disease modeling of adult-stage conditions. Several aspects regarding cardiomyocyte maturation, such as the necessary time to develop the disease phenotype, need to be addressed in order to obtain robust *in vitro* models for AFib. Different approaches have been suggested to boost the maturation of hPSC-derived cardiomyocytes by identifying the missing factors characteristic of adult cardiomyocytes. Mainly, cardiomyocytes should recapitulate the expression of specific genes that are representative of the adult stage, the exhibition of an organized cytoskeleton and specialized organelles, an oxidative metabolism, and an adult electrophysiological behavior. Kolanowski et al. reported strategies to stimulate maturation considering these features, such as biochemical stimulation (by activation or repression of signaling pathways), physical stimulation (through electrical or mechanical stimuli) as well as the development of 3D models able to reinforce the mechanical properties of cardiomyocytes [133]. Moreover, cardiomyocytes *in vivo* strongly interact with non-cardiomyocytes, influencing cardiac development and specification. For instance, cardiac fibroblasts play an important role by providing the extracellular matrix of the heart, mechanical support, and paracrine factors. Cardiac endothelial cells contribute to myocardial microvasculature, regulating oxygen and fatty acids supply to cardiomyocytes. Therefore, the combination of these different cardiac populations at controlled rates might improve the maturity of hPSCs-cardiomyocytes [134]. Nevertheless, there is no current protocol that combines all different approaches representing the *in vivo* cardiac physiology (macro- and microenvironments) exploring the full potential of hPSC-cardiomyocytes to be used for high-throughput screening and therapeutic applications for AFib. It is important to highlight the progress provided by the advent of CRISPR/Cas9 technology in the development of disease model lines of AFib harboring different genetic variants known to have AFib-related effects. Incorporation of these disease lines and genetic tools can provide a great advantage for a drug screening platform. In addition, although genetically modified disease lines replicate the genetic component of AFib pathophysiology, non-genetic factors also need to be taken into account when modeling arrhythmia. For that, more advanced 3D-defined multicellular models combining different AFib-leading features are required. Illustrative of that would be a geometrically-constrained culture of cardiomyocytes with cardiac fibroblasts, endothelial and smooth muscle cells as a way of developing engineered atrial heart tissue capable of modeling the electrophysiological mechanisms characteristics of AFib through environmental elements that contribute to the initiation, maintenance and progression of AFib. In this context, 3D (bio)printing and advanced bioengineering techniques can serve as a promising strategy for further developing such models, by enabling precise spatial arrangement of each cell type.

In conclusion, taking into account most available models and the drawbacks associated with each one (Table 2), it is clear that the complex multifactorial phenotype of AFib can only be fully appreciated in a 3D anatomically comprehensive system, and perhaps a simple monolayer stem cell model will not be able to capture structural aspects of the substrate that are important in AFib pathogenesis. Hence, it is unrealistic to expect a single model to fully capture all aspects of AFib. Instead, the integration of diverse models, including *in vivo*, in vitro, and computational approaches, can enhance our understanding of the molecular mechanisms driving the onset, progression, and persistence of the disease. This comprehensive approach ultimately paves the way for novel personalized therapeutic interventions tailored to individual patients.

## Figures and Tables

**Figure 1 biomedicines-11-02355-f001:**
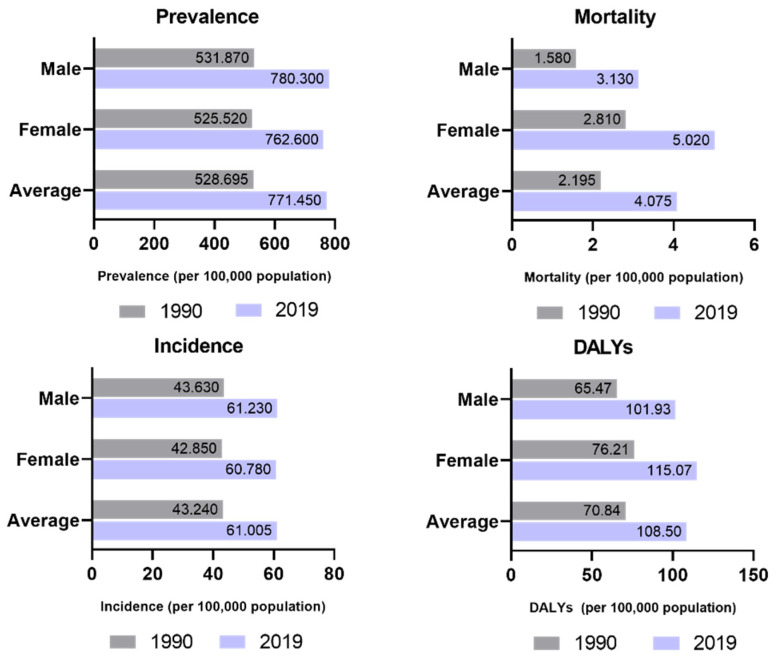
Epidemiology of AFib. Strong increase in prevalence, incidence, mortality, and disease burden from 1990 to 2019.

**Figure 2 biomedicines-11-02355-f002:**
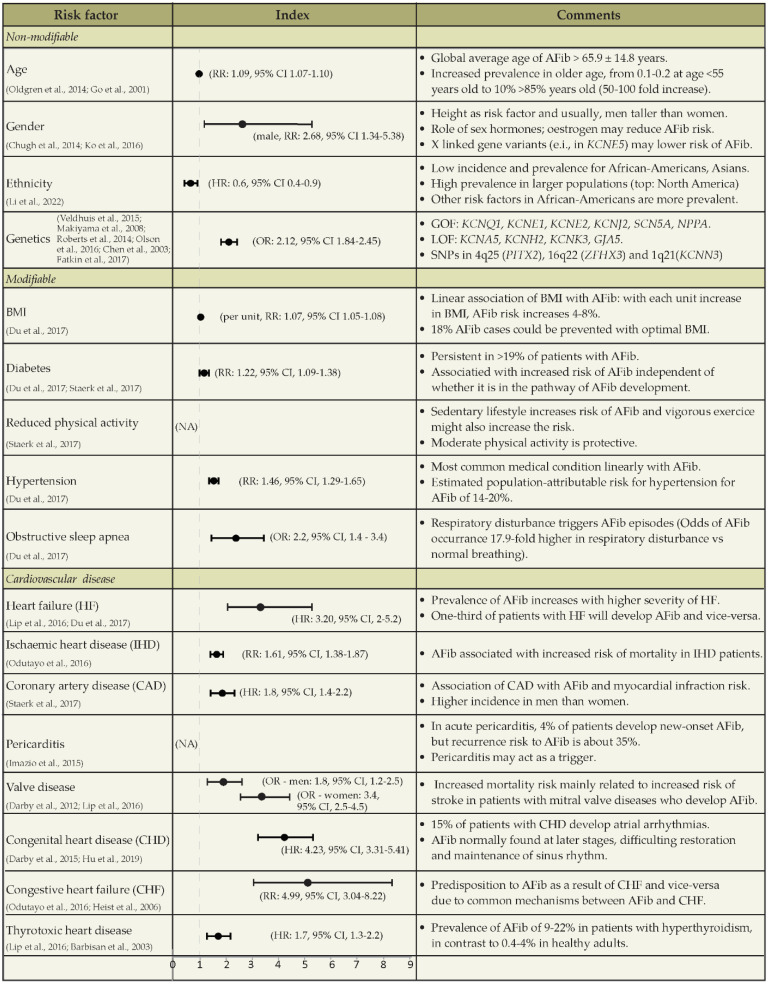
Risk factors for AFib. AFib, atrial fibrillation; GOF, gain-of-function mutation; LOF, loss-of-function mutation; *KCNH2*, gene encoding a subunit of the voltage-gated potassium channel K_v_11; *KCNK3*, gene encoding TASK-1, a two-pore domain K^+^ channel with atrial-specific expression shown to be a major determinant of resting membrane potential in human atrial cardiomyocytes; *GJA5*, gene encoding for connexin 40; *NPPA*, gene encoding the precursor protein for atrial natriuretic peptide (ANP), a circulating hormone produced in the cardiac atria; BMI, body mass index; NA, not applicable; RR, relative risk; HR, hazard ratio; OR, odds ratio; CI, Confidence interval; RR, Relative Risk; HR, Hazard Ratio [2,5,11,12,13,14,15,16,17,18,19,20,21,22,23,24,25,26,27,28].

**Figure 3 biomedicines-11-02355-f003:**
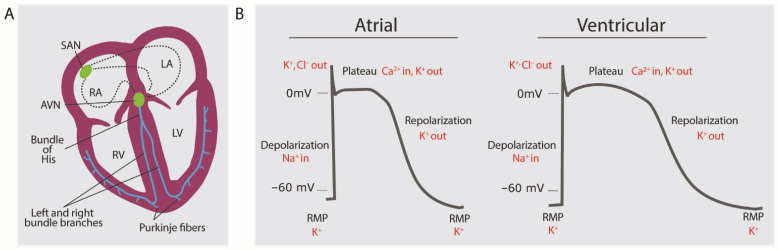
(**A**) Propagation of electrical signals within the heart. (**B**) Atrial and ventricular action potential in healthy cardiomyocytes. SAN = Sinoatrial node; AVN = Atrioventricular node; RA = Right atrium; LA = Left atrium; RV = Right ventricle; LV = Left ventricle; RMP = Resting membrane potential. The dashed line in A indicates electrical propagation.

**Figure 4 biomedicines-11-02355-f004:**
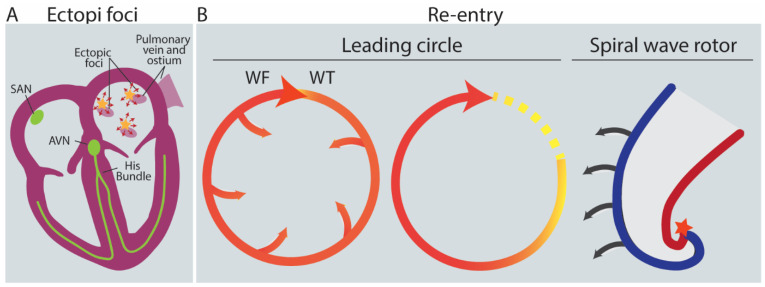
Types of triggers for AFib. (**A**) Rapid firing ectopic foci, for example arise in the pulmonary vein. (**B**) Types of re-entry. Left: re-entrant circuit around an anatomical barrier. Depending on the size of the excitable gap (space between WF and WT), re-entry will be sustained or not. Middle: a leading circle with no excitable gap (WF anchors WT) forming a refractory area in its center due to continuous centripetal activation. The smallest circuit that can sustain re-entry. Right: spiral wave re-entrant rotor. At the singularity point (star), an excitable but not excited core is formed. (Depending on the curvature of the WF blue), the rotor will be maintained. Arrows indicate velocity. SAN = Sinoatrial node; AVN = Atrioventricular node; WF = Wavefront; WT = Wavetail; WL = Wavelength [3,49,50,51].

**Figure 5 biomedicines-11-02355-f005:**
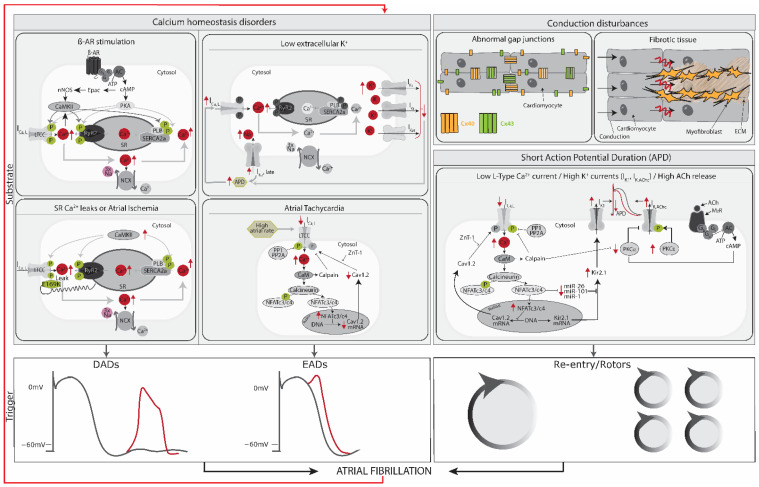
Overview of underlying disease mechanisms of AFib. Calcium disorders resultant from β-adrenergic (β-AR) stimulation, sarcoplasmic reticulum (SR) Ca^2+^ leaks, or atrial ischemia typically result in diastolic Ca^2+^ release from the SR, increasing cytosolic Ca^2+^ and activating the NCX exchanger which produces a transient-inward current causing a membrane depolarization (Delayed AP Duration, DAD—red line) [3,41,52]. In contrast, calcium disorders, such as low extracellular potassium (K^+^), allow L-type Ca^2+^ (I_Ca,L_) channels to recover from the inactivation that occurs during the AP, generating an inward current and resulting in Early APDs (EADs—red line) [3]. Atrial tachycardia can also act as a substrate for EADs, caused by a reduced L-type Ca^2+^ channel (I_Ca,L_) and the increase in inward-rectifier currents (I_K1_ and I_KACh,C_), overall reducing the APD [3,53,54]. Conduction disturbances, such as abnormal gap junctions or tissue fibrosis, and short APDs act as substrates for re-entry and rotors [2,55]. The triggers ultimately lead to AFib. With AFib progression, substrates might change, reinforcing the vulnerability of the substrate to maintain AFib (“AFib begets AFib,” looping red line from trigger to substrates). Red arrows up: increase; red arrows down: decrease. Black line in DADs/EADs represents normal reference action potential and the red line the DAD or EAD tracing. APD = Action potential duration. SR = Sarcoplasmic reticulum. E169K = Junctophilin-2 (JPH2) missense mutation. Cx = connexin. Ach = acetylcholine. M_2_R = M2 muscarinic receptor.

**Figure 6 biomedicines-11-02355-f006:**
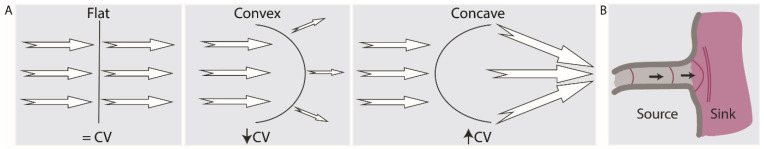
(**A**) Effect of wavefront curvature on conduction velocity (CV) [57,60]. (**B**) Current-to-load mismatch where the source volume is smaller than the sink volume, leading to a block of the activation wavefront (purple lines). Arrows indicate direction of electrical propagation.

**Table 2 biomedicines-11-02355-t002:** Models of AFib with their advantages (pros) and disadvantages (cons). AADs = Anti-arrhythmic drugs.

	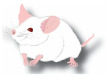 *In vivo* models	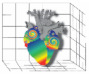 *In silico* models	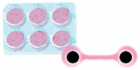 *In vitro* models
PROS	Full access to tissues and cells [53]Careful control of variables [50]Study of single cause-effect relationship [50]	Multiscaled integrated models [105]Controlled ionic and molecular inputs [105]Evaluation of therapy efficacy and patient-specific structural remodeling [105]	Mimic human physiology [135]Single-cell or mini-tissue level [33]Capture health and disease phenotypes [33]Screen for AADs [106]High throughput [135]Disease lines of AFib by CRISPR/Cas9 [132]
CONS	Interspecies differences and variability [114]Ethical issues [114]Limited availability of spontaneous-AFib models [136]	Reliability on data from other animal models [88]Lack of information of arrythmias maintenance [105]Lack of systemic influences [135]	Need for 3D anatomically comprehensive system [135]Isolation of specific cell subtypes [114]Immature phenotype [114]Lack of systemic influences [135]

## Data Availability

Not applicable.

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
