# Peer review of "Towards Improved Human In Vitro Models for Cardiac Arrhythmia: Disease Mechanisms, Treatment, and Models of Atrial Fibrillation"

_biomedicines, 2023, doi:10.3390/biomedicines11092355_

Round 1
Reviewer 1 Report
This review effectively presents the current findings on cardiac arrhythmia, a disorder that disrupts the normal heart rhythm. Additionally, it critically assesses the current cardiac arrhythmia models, while also shedding light on potential avenues for further refinement. It particularly delves into atrial fibrillation, a prevalent form of cardiac arrhythmia characterized by disturbances in heart's electrical communication, resulting in impaired synchronization and function. Overall, this review is well-written and comprehensive. However, the following concerns need to be addressed before publication.
1. To enhance the impact of the review, consider including a figure in the "Brief overview of normal electrophysiology of cardiomyocytes" section. This visual aid could illustrate the propagation of electrical signals within the heart, facilitating better comprehension for readers.
2. The review comprehensively addresses the topic, but there's an opportunity to enhance its clarity further. It would be beneficial for the author to re-evaluate each section for extraneous information. Some sections tend to be overly extensive, and isolated pieces of information might divert readers from the central theme. Streamlining the content will ensure that readers remain focused on the key points throughout the review.
3. In order to maximize the usefulness of figures in the paper, the author could consider enlarging figures 4 and 5. Currently, some details are not easily visible, especially the pathways in figure 4 and the text in figure 5.
Author Response
Dear reviewer, Thank you for the positive feedback. Your comments are very worthwhile and we have addressed them to improve our review manuscript.
- To enhance the impact of the review, consider including a figure in the "Brief overview of normal electrophysiology of cardiomyocytes" section. This visual aid could illustrate the propagation of electrical signals within the heart, facilitating better comprehension for readers.
Thank you for this suggestion, we have now included a figure about the cardiac conduction pathway and the cardiac action potential in healthy atrial and ventricular cardiomyocytes in figure 2 of the revised manuscript.
Figure 2. A) Propagation of electrical signals within the heart. B) Atrial and ventricular action potential in healthy cardiomyocytes. SAN = Sinoatrial node; AVN = Atrioventricular node; RA = Right atrium; LA = Left atrium; RV = Right ventricle; LV = Left ventricle; RMP = Resting membrane potential. Dashed line in A indicates electrical propagation.
- The review comprehensively addresses the topic, but there's an opportunity to enhance its clarity further. It would be beneficial for the author to re-evaluate each section for extraneous information. Some sections tend to be overly extensive, and isolated pieces of information might divert readers from the central theme. Streamlining the content will ensure that readers remain focused on the key points throughout the review.
Thank you for this suggestion. We have scanned the manuscript again and removed isolated sentences and generated a separate paragraph for the in silico models to streamline the conclusion and future perspectives section.
- In order to maximize the usefulness of figures in the paper, the author could consider enlarging figures 4 and 5. Currently, some details are not easily visible, especially the pathways in figure 4 and the text in figure 5.
We have now changed figure 5 into a table to increase readability of the text. We also increased the size of the pathways in figure 4.
Reviewer 2 Report
well done review....summing a lot of sources.
"Heart rhythm disorders, so-called arrhythmias" sounds.... tricky. please delete "so-called"
Author Response
Dear reviewer, Thank you for the positive comment and your effort to review our manuscript. We have now deleted "so-called".
Reviewer 3 Report
This is an interesting and up-to-date review on pathogenetic mechanisms of atrial fibrillation and different models of cardiac arrhythmia.
Suggestions for the authors:
- Please revise the title to better cover the actual content of the review. Indeed, this review does not focus exclusively on disease models and the main conclusions and future perspectives are that in vitro models unlikely can recapitulate all relevant features of atrial fibrillation to serve as models for the development of new therapeutic approaches.
- References are not cited in the text according to the journal guidelines. References should be numbered in order of appearance and cited as numbers in brackets throughout the text.
- Sections and subsections need to be numbered.
- References in Tables 1 and 2 should be properly inserted in an additional column on the right. Table 1 should be better formatted. Otherwise, I suggest to replace Table 1 with a figure.
- Please explain whether Figure 3 or some of the figure parts are original or not. Indeed, there are some references in the legend.
- Legend of Figure 4 should be more explicative and comprehensive.
- What is shown in Figure 5 would be much more explicative if presented as table with appropriate references.
- Please check the text for typos. Moreover, be consistent throughout the text in using only US or British english (e.g., modeling/modelling).
Please check the text for typos. Moreover, be consistent throughout the text in using only US or British english (e.g., modeling/modelling).
Author Response
Dear reviewer, Thank you for the positive comment. We appreciate your effort to review our manuscript and have addressed your comments in the revised manuscript.
Suggestions for the authors:
- Please revise the title to better cover the actual content of the review. Indeed, this review does not focus exclusively on disease models and the main conclusions and future perspectives are that in vitro models unlikely can recapitulate all relevant features of atrial fibrillation to serve as models for the development of new therapeutic approaches.
We agree with the reviewer that the title could be more reflective for the content of this review. We have now changed the title to: Towards improved human in vitro models for cardiac arrhythmia: Disease mechanisms, treatment and models of atrial fibrillation.
- References are not cited in the text according to the journal guidelines. References should be numbered in order of appearance and cited as numbers in brackets throughout the text.
Thank you for noticing this error. We have now changed the reference format to the one of the journal.
- Sections and subsections need to be numbered.
Thank you for this suggestion, we have now included the right numbering.
- References in Tables 1 and 2 should be properly inserted in an additional column on the right. Table 1 should be better formatted. Otherwise, I suggest to replace Table 1 with a figure.
Thank you for the suggestion. We have included the references in each table. We also formatted Table 1 to improve readability.
- Please explain whether Figure 3 or some of the figure parts are original or not. Indeed, there are some references in the legend.
Dear reviewer, panel A of this figure (now Figure 4 in the improved manuscript) has been adapted from the picture by V. G. Fast and A. G. Kléber, to more clearly show the effect on the conduction velocity by adapting arrow size. We included the references in the figure legend. Panel B of this figure is original.
- Legend of Figure 4 should be more explicative and comprehensive.
We agree and we have now provided more information in the legend of Figure 4 (note that now is Figure 5).
- What is shown in Figure 5 would be much more explicative if presented as table with appropriate references.
We agree with the reviewer and accordingly changed the figure into a Table (Table 3), with the appropriate references for each statement.
- Please check the text for typos. Moreover, be consistent throughout the text in using only US or British english (e.g., modeling/modelling).
We have now changed the manuscript text to be more consistent with US English.